# The Impact of Sustainable Tourism on Resident and Visitor Satisfaction—The Case of the Special Nature Reserve "Titelski Breg", Vojvodina

**Igor Trišić** [1,*], **Snežana Štetić** [2,3], **Adina Nicoleta Candrea** [4], **Florin Nechita** [5], **Manuela Apetrei** [6,*], **Marko Pavlović** [7], **Tijana Stojanović** [8] **and Marija Perić** [7]

1   Faculty of Geography, University of Belgrade, Studentski Trg 3/III, 11000 Belgrade, Serbia
2   International Research Academy of Science and Art, Kašikovićeva 1a, 11010 Belgrade, Serbia; snezana.stetic@gmail.com
3   Balkan Network of Tourism Experts, 11000 Belgrade, Serbia
4   Faculty of Economic Sciences and Business Administration, Transilvania University of Brașov, 500036 Brașov, Romania; adina.candrea@unitbv.ro
5   Department of Social and Communication Sciences, Transilvania University of Brașov, 500036 Brașov, Romania; florin.nechita@unitbv.ro
6   Centre for Mountain Economy, "Costin C. Kirițescu" National Institute for Economic Research, Romanian Academy, 060021 Bucharest, Romania
7   Academy of Technical Vocational Studies in Belgrade, Katarine Ambrozić 3, 11120 Belgrade, Serbia; markopavlovic25101982@gmail.com (M.P.); mperic@atssb.edu.rs (M.P.)
8   Faculty of Applied Ecology "Futura", Metropolitan University, Požeška 83, 11030 Belgrade, Serbia; tijana.stojanovic@futura.edu.rs
*   Correspondence: trisici@hotmail.com (I.T.); manuela.apetrei@ce-mont.ro (M.A.); Tel.: +381-64-143-13-75 (I.T.); +40-756-140-147 (M.A.)

**Abstract:** The Special Nature Reserve "Titelski Breg" (TB) is a protected area (PA) of category I, located in the AP of Vojvodina in the south-eastern part of Bačka. The reserve covers an area of 496 ha. A protection zone covering an area of 8643 ha has been established around the PA. The International Union for Conservation of Nature (IUCN) states that this PA is classified as a category IV habitat and species management area. Its good geographical and traffic position and close distance to Romania and Hungary, as well as the nation's major cities, make this PA accessible to a sizable number of both domestic and foreign tourists. There are numerous plant and animal species in the reserve, which makes this area unique. The population living around this reserve has an exceptional and valuable cultural heritage, which represents significant complementary tourist motives. To examine the state of sustainable tourism (SuT) in TB and the impact of SuT on the satisfaction of the respondents (SoR), the PoS model of study was used. The quantitative methodology in this research included a questionnaire as the survey instrument for residents and visitors. There were 630 respondents altogether (400 locals and 230 guests). Four aspects of sustainability, economic, social, cultural, and institutional, were used to analyze the state of SuT in this PA. The study's findings show that SuT significantly affected the SoR. Analyzing the role that additional protected areas may have in SuT can be supported by the research outcomes. Additionally, the proportion of each sustainability characteristic in SuT can suggest guidelines for national programs that aim to develop PAs and tourist development at the same time.

**Keywords:** resident and visitor satisfaction; special nature reserve; prism of sustainability; protected natural asset

## 1. Introduction

The Special Nature Reserve "Titelski Breg" (TB) is a protected area (PA) of category I, located in the AP of Vojvodina in the south-eastern part of Bačka. The Special Nature

Reserve (SnR) covers an area of 496 ha. A protection zone covering an area of 8643 ha has been established around the PA [1,2].

The main characteristic and value of this PA are the special geological form and relief of its loess section. It is a loess plain that was created by the deposition of loess during the Pleistocene. Like other priceless natural rarities, this type of relief is an aspect of geoheritage that must be conserved due to its uniqueness. The loess section in TB has multiple values and functions. These are scientific and educational value, cultural and social value, geodiversity value, and visual and aesthetic value [3]. Due to the specific values that TB possesses, the development of tourism can be based on the foundations of geotourism [4]. The proper development of tourism in TB can preserve and improve the values that this PA possesses. The subject of research in this paper is the effects of SuT on TB and its influence on the satisfaction of the respondents (SoR). The state of sustainable tourism (SuT) and sustainable tourism development (SuTD) was examined according to four components: economic, social, cultural, and institutional. The PoS research model is based on this. In addition to the above, the research examined the specific effects of the sustainability dimensions on tourism development (TD) and the SoR. The results of examining the impact of tourism on satisfaction with SuT among residents and visitors can provide significant scientific information [5,6]. This information can be used for action measures to identify, evaluate, and implement [7] underdeveloped or unused natural and social values, important for the advancement of SuT [8–10]. Additionally, the research's findings may be helpful in the creation of local, regional, and national TD strategies that include PAs in the tourist offering [11–13].

The aim of this paper is to determine the degree and condition of SuT and its importance to this SnR using the obtained research results, that is, to determine the level of the impact of SuT on the SoR. The research gap is precisely the state of sustainable tourism in TB. It is necessary to examine the extent to which the state of tourism development and perspectives on TD can contribute to sustainability without jeopardizing the important geological features and values of this PA. Therefore, the main research questions concern the examination of the current situation of sustainable tourism and its impact on the satisfaction of the users of this space. The used research model allows us to collect certain information regarding the relationship between nature and tourism. The main research questions relate to determining the state and impact of SuT. This can be discovered by looking at each of the four sustainability pillars separately, as the description of the study model suggests. One of the research issues is which of the four characteristics of sustainability makes the biggest contribution to sustainability. Furthermore, it is essential to examine the degree of satisfaction that PA users have with the current condition of tourism.

The level of SoR, who are users of this PA, can indicate the state and degree of development of SuT [14], as well as the importance of the development of specific forms of tourism [15]. In addition to the above, the goal of this research is to examine whether TB can be a destination for nature-based ecotourism, that is, whether the TB can be a destination for SuT.

A quantitative methodology was used for the purpose of this paper. A written questionnaire was used as the instrument for the survey, which included 630 respondents in total (400 locals and 230 guests). The respondents were selected using a random sampling method. The obtained data were statistically processed with the help of SPSS v.21 software and presented in tabular form. The statistical method included Cronbach's alpha coefficient and simple linear regression.

The research's findings may be crucial to the advancement of local, national, and regional TD strategies. The research will identify stronger and weaker factors of TD in TB. The expressed attitudes of the interviewees may indicate the importance of certain dimensions of sustainability to SuT. By analyzing individual values, proposals for the development of specific forms of tourism of a sustainable character can be made. Finally, the results of this research can indicate whether PAs can be significant destinations for SuT.

One form of research limitation was the impossibility of contacting residents in the spring and early summer months due to bad weather conditions. The residents were unwilling to participate in the survey because of the tangible harm that powerful winds had caused to their property. As a result, the focus on establishing personal contacts was moved to the fall, extending the duration of the field data gathering process.

An obstacle in the research was ensuring the competencies of the respondents when filling out the questionnaire and answering the questions. Therefore, the questionnaire had to contain significant instructions for filling it out.

## 2. Literature Review

There are 138 PAs in the territory of the AP of Vojvodina in total, covering an area of 148,599.6 ha, which occupies 6.91% of the land of the province. SnRs make up a significant share of the area of the PAs (a total of 16 SnRs) [1,2]. The SnRs of Vojvodina differ in terms of their relief and soil composition (mountains, plains, loess sections, dunes, alluvial plains), territorial distribution (larger and smaller coverage of the territory), the diversity of their flora and fauna, endangered species status, wetlands, etc. [16–18].

In recent years, SuTD has included examining the development of tourism in PAs due to increasing anthropogenic impacts on ecology [19,20]. In PAs, nature is the primary resource that needs to be preserved [21,22]. The major negative impacts of the growth of tourism on PAs are changes in geographical conditions and the influence on flora and fauna [23,24]. All tourist activities in PAs must be harmonized with the protection of nature and the promotion of its standards [25,26]. Apart from the ecological principles, the management of these protected destinations includes achieving a positive sociocultural and economic climate and conditions for the development of SuT [17].

TB has special natural features in terms of its hydrography, relief, geological forms, and ecosystems, which make it unique in this part of the province [27]. The people that live in the vicinity of this PA are from several ethnic groups and have rich cultural traditions. Among the most important involve their cultural values, heritage, historical heritage, customs, gastronomy, local handicrafts, and local events [28]. Social characteristics can be important complementary tourism motives, which, together with primary natural motives, can create a SuT destination [29,30].

The inclusion of residents in the planning of TD, realizing interactions between locals and guests, and enhancing their involvement in planning and development processes will benefit this PA in terms of the environment, economy, society, and institutions [31–33].

Properly planned TD could also contribute to local economic development. The positive impacts of TD include an increase in jobs for residents [34–36], the strengthening of local culture and local crafts, the development of infrastructure, the development of nature-based forms of tourism, ecotourism, birdwatching, trips, science tourism, etc. [37,38]. The most important positive impacts of TD are certainly improvements in natural values [39] and the protection of the nature of this reserve [40]. Ensuring material gains through the development of tourism enables more significant investment in the protection of this reserve and its natural values [41–43].

Aktymbayeva et al.'s [44] research considered the interrelation between SuT and environmental conservation, as this has become a major concern in contemporary research, motivated by the need to harmonize the economic benefits of tourism with ecological preservation. They demonstrated that the carrying capacity of a specific park in Kazakhstan is not a static figure but a variable that requires constant recalibration, reflecting the park's ecological health and visitor perceptions. The results of the study indicate that the carrying capacity and the limits of acceptable changes affect a reduction in satisfaction due to over-tourism. Sustainability can be brought back to normal with the help of urgent and adequate management measures in this area.

The study model was designed by the authors using the following four studies as a guide. It was tailored to collecting data on the status of sustainable tourism and its effect on respondents' satisfaction:

(1) Huayhuaca et al. [45] examined four dimensions of sustainability using respondents' perceptions of the state of SuT in Frankenwald Nature Park. Residents ranked their responses to 21 items on a 7-point Likert scale of agreement. The items were grouped into four dimensions. The ecological dimension of sustainability was rated by the respondents as the most important dimension. Planning the development of tourism in this PA should be in accordance with ecological principles. The carrying capacity, zoning, the application of ethical codes, and tourism infrastructure development stand out here. Also, the economic and sociocultural sustainability aspects have been found to be crucial to the advancement of SuT. The findings of this study offer crucial details and recommendations for the improvement of management for the growth of tourism. By examining the four dimensions of sustainability, it is possible to identify stronger and weaker factors that enable tourism planning in a PA. The Prism of Sustainability model was used in the research, which led to adequate identification of the positive and negative aspects of the growth of tourism. In order for this process to be successful, SuT was observed according to the four pillars of sustainability, as the model was conceived. This research model was used by the authors in designing the research model in this article.

(2) Stojanović et al. [46] examined the function of PAs in SuTD. The research was based on the PoS methodology. Wetland was chosen for the research area, which offers the possibility of developing nature-based forms of tourism. The research was compiled to examine the importance of natural and sociocultural values to the development of tourism. The respondents stated that natural factors were crucial to nature-based tourism and ecotourism. The local community that inhabits the area around the reserve has a significant cultural heritage. These sociocultural values can complement tourist activities. By combining the natural and sociocultural values of this SnR, a tourist offering can be formed, which can significantly contribute to SuTD.

(3) Shen and Cottrell [47] examined residents' satisfaction with agritourism in four agritourism villages in China. Four aspects of sustainability were examined with the help of 3 to 5 research questions. The results of the research indicate the presence of significant satisfaction in the residents. All the dimensions of sustainability had a significant contribution to satisfaction. Institutional sustainability was the most important, followed by economic and sociocultural, with a share of 80% in the total satisfaction of the residents. The concluding considerations indicated the need to include all four dimensions of sustainability when monitoring the sustainable development of agritourism.

(4) Trišić et al. [48] examined the importance of the UNESCO Biosphere Reserve "Mura-Drava-Danube" for the development of sustainable tourism. The sample consisted of residents (1295) who lived in the area around the reserve in three border countries: Serbia, Croatia, and Hungary. Surveys were used to collect data that indicated that environmental sustainability, as an important dimension of sustainability, has the greatest impact on SuT. It is obvious that the sociocultural and institutional dimensions have a smaller share in the impact of sustainable tourism on the satisfaction of respondents. Greater employment of locals and the economic effects of tourism development will be more significant after the proper expansion of nature-based and cultural forms of tourism. One of the results of the research indicates that this reserve can be a sustainable tourism destination with the proper development of specific forms of tourism and with the involvement of locals in the planning and development processes. The results were relatively similar for all three border countries. Such a result can help in the preparation of international planning documents concerning cross-border cooperation in the spread of sustainable forms of tourism.

## 3. Research Area

The integral part of TB is in the AP of Vojvodina, in south-eastern Bačka. This SnR stretches from 45°12′19″ to 45°17′50″ N and from 20°07′53″ to 20°18′58″ E. With an area of 496 ha, it covers the territories of Titel, Mošorin, Lok, and Vilovo [27]. A protective zone covering an area of 8643 ha has been established around the PA. Protection zones of the first degree (13.94%), second degree (49.62%), and third degree of protection (36.44%) were

established within the reserve. This reserve acquired national protection status in 2012 as a category I PA. The International Union for Conservation of Nature (IUCN) has placed this PA in category IV as a habitat and species management area. Due to its botanical characteristics, and according to the IPA criteria (Important Plant Areas), the floristic and ecosystem characteristics of this PA served as the basis for its inclusion in the IPA areas of Central and Eastern Europe. This area is inhabited by many different species of avifauna. Some species are extremely rare and endemic. Based on the previously described factors, TB was included on the Important Bird Areas (IBA) list of significant bird areas.

The area of the reserve consists of unique geological loess forms (loess plateaus). In addition, the reserve is characterized by valleys, loess pyramids, hanging valleys, chasms, shoulders, and surducs, as well as dual forms of fluvial/karst erosion, that are over 600 years old [49,50]. This form of relief was created 6000 years ago [51]. The significant natural hydrographic potential of this PA is represented by the proximity of the Begej, Tisa, and Danube rivers. At the confluence of the Tisa and the Danube, rare heathy and alluvial soil was formed, characterized by numerous wetlands, important to the ecosystem of this SnR. Figure 1 shows the location of TB in relation to the nation's and the region's larger cities.

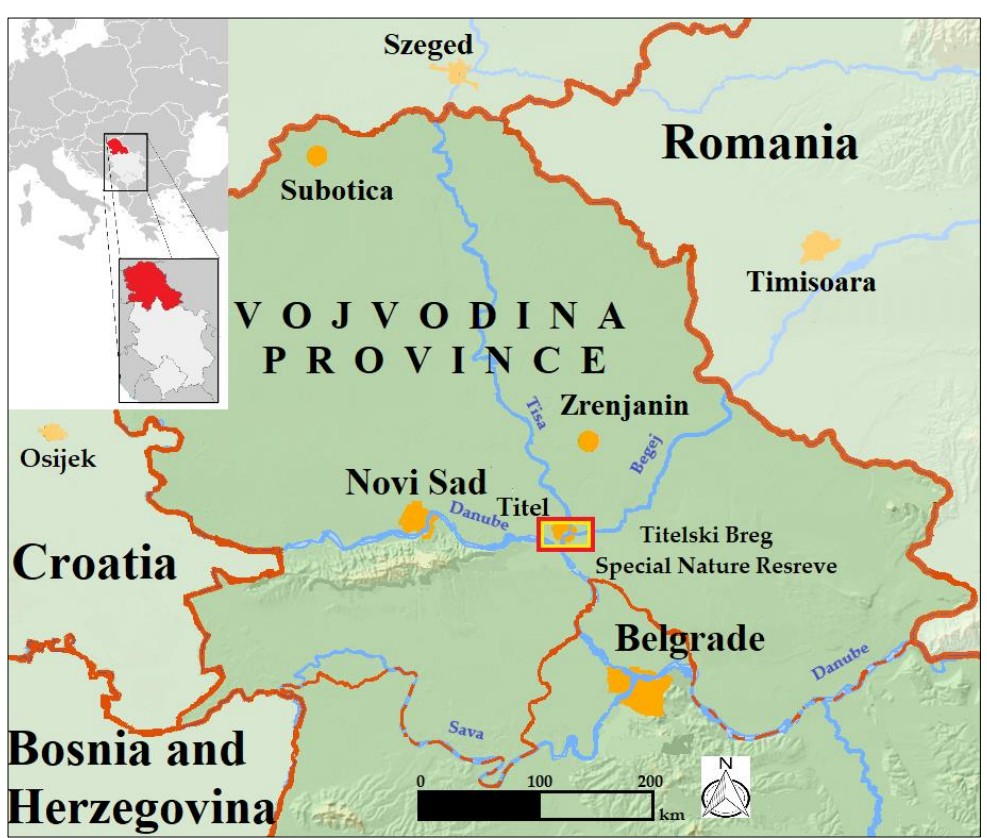

**Figure 1.** Study area. Source: Trišić, I., author.

About 630 plant species, 137 bird species, 9 amphibian species, 11 reptile species, and 33 mammal species rare in this area were recorded in TB. Among the most important representatives of flora, the following should be highlighted: *Sternbergia colchiciflora*, *Adonis vernalis*, *Alyssum linifolium*, *Alkanna tinctoria*, *Prunus tenella*, *Allium rotundum subsp. waldsteinii*, *Crocus variegatus*, *Bassia sedoides*, *Sysimbrium polymorphum*, *Iris pumila*, *Pulsatilla pratensis,* and others.

Representatives of the fauna that characterize this SnR are Coracias garrulus, Aquila heliaca, Falco cherrug, Falco tinnunculus, Upupa epops, Emys orbicularis, Zamenis longissimus, Lepus europaeus, Neomys fodiens, Spermophilus citellus, Clethryonomys glareolus, Micromys minutus, Vulpes, Mustela nivalis, Martes foina, Martes foina, Felis silvestris, and

other significant species [27]. The loess plateaus are a special feature of this PA, which are used for the nests of Riparia riparia and Merops apiaster, as can be seen in Figure 2.

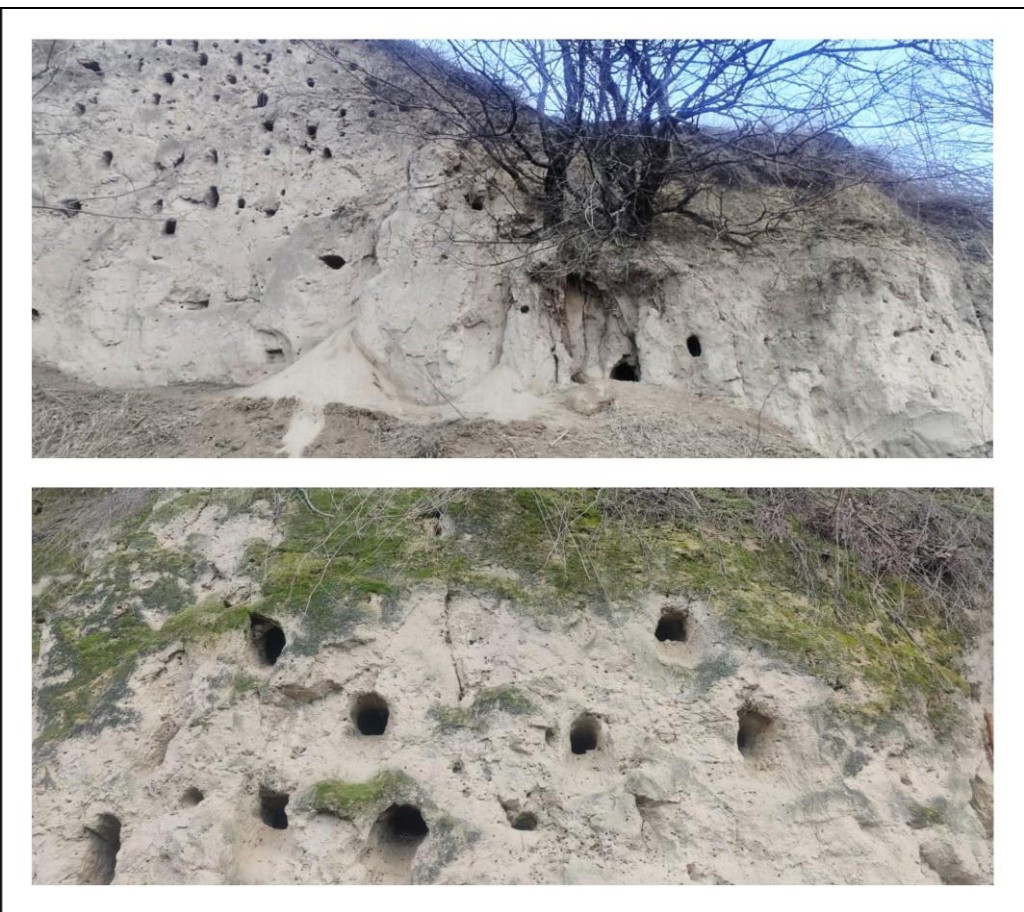

**Figure 2.** Nests in loess plateaus. Source: Trišić, I., author.

## 4. Methodology

### 4.1. Procedure

This research represents the authors' continuation of the examination of the role that the PAs of Vojvodina have in SuTD. The inclusion of TB in the research area is significant in that it facilitates obtaining more reliable scientific results about the state of SuT and the role that the PAs of Vojvodina have in SuTD. Involving as many respondents as possible in the study process also helps to produce more trustworthy scientific findings and conclusions. The research was conducted using the random sampling method. The respondents were surveyed via personal contact, through visits to the PA, and through thematic social networks (via e-mail and online questionnaires). Surveying the respondents was carried out from May to October 2023.

### 4.2. Instruments

This research is based on a quantitative methodology, which included a survey of the respondents as a research technique. The Prism of Sustainability study model (PoS) was applied in the present study. The model of research was designed according to the study of SuT in PAs as important tourism destinations [43,45,52]. The questionnaire was adapted to collect the responses and attitudes of residents and visitors according to certain items [45]. The aim of this adapted PoS model is to examine the state of SuT in this PA through the four dimensions of sustainability, as well as its impact on the SoR [5,31]. The environmental, economic, social, and institutional components are the four sustainability

aspects of concern [43,45]. This research differs from previous research in terms of the structure of the respondents. The target groups in the survey were residents and visitors. A comparative analysis of the obtained values for both groups of respondents can lead to more reliable results. The research model can be seen in Figure 3.

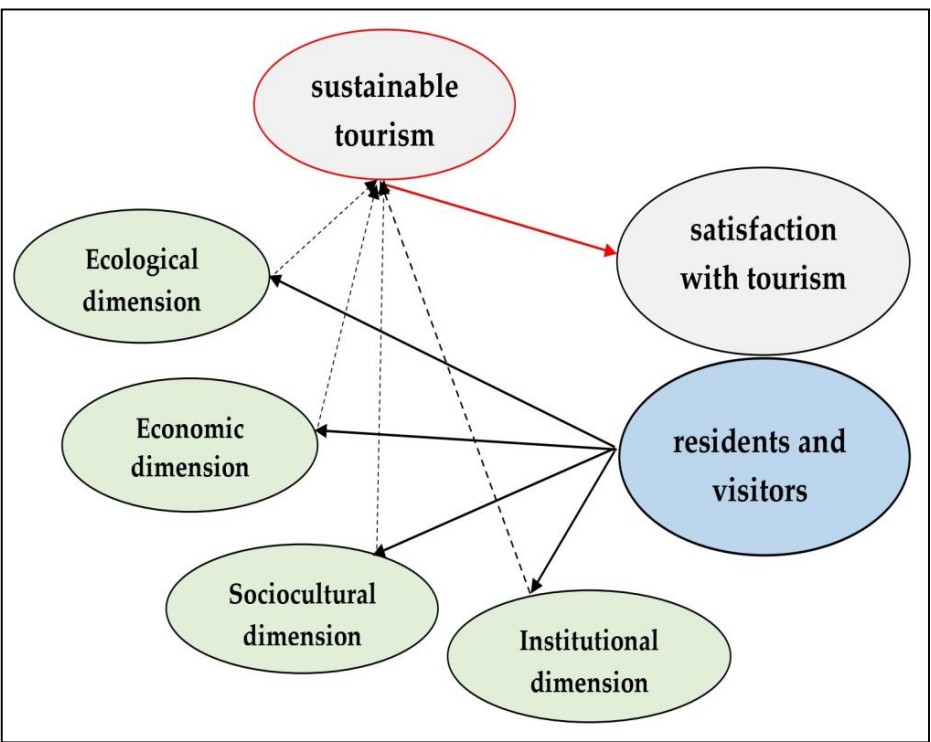

**Figure 3.** The conceptual model. Source: Trišić, I., author.

The respondents filled in the questionnaire completely anonymously. The respondents granted permission for the research findings to be used for scientific purposes and for the publication of the scientific results by completing the questionnaire.

*4.3. Data Analysis*

The reliability of the responses provided was examined using Cronbach's alpha coefficient during the statistical processing of the data. Cortina [53] and Nunnally and Bernstein [54] suggest that an alpha value of less than 0.60 can be accepted in research studies. In addition, each completed questionnaire was controlled. The confirmatory factor analysis (CFA) method of checking the reliability of the variables was used in the research as a supplementary method for checking the obtained values for all the dimensions of sustainability. Also, the same method was used to check the respondents' satisfaction with SuT. The impact of the sustainability dimensions on the SoR was examined using a simple regression analysis [55,56]. The investigated institutional indicators are the importance of legal legislation to the protection of the area, the importance of management processes, the existence of guiding and educational centers, the possibility of developing scientific forms of tourism, and others [44,45]. The ecologically tested indicators refer to the attitude of the users of the space towards the ecology and protection of the area. Among the most significant examined indicators of the ecological dimension of sustainability are the endangerment of flora and fauna, the creation of tourism facilities and infrastructure to provide services for tackling tourist pollution, and the exploitation of resources from the reserve [45]. The examined sociocultural indicators refer to the local population's contribution to education and the development of tourism and the promotion of local culture, as well as the interaction of residents and visitors. Economic indicators refer to benefits of the development of sustainable forms of tourism [44–48].

## 5. Results

The random sampling approach was used to poll 630 people in total (400 residents and 230 guests). Following a thorough process of determining each questionnaire's validity separately, the number 630 denotes the number of approved questionnaires. A total of 37 questionnaires were invalid for analysis. A total of 460 respondents were surveyed with the help of an online questionnaire (327 residents and 133 visitors), while 170 respondents were surveyed using personal contact (73 residents and 97 visitors). The settlements in which the residents were surveyed were Titel (64%), Mošorin (27%), and Vilovo (9%). Visitors from within the country made up 79% of the total respondents. The countries from which foreign visitors originated were Hungary (34%), Romania (18%), Montenegro (12%), Croatia (10%), North Macedonia (9%), Austria (8%), Switzerland (4%), and other countries (5%). The majority of respondents (58%) of the total were women. The average age for both groups of respondents was 33 (from 18 to 81). Most of the respondents had secondary education, 57%, a total of 21% had primary education, and 20% had a college or university degree, while 2% of respondents had a master's or doctorate degree.

The analysis of the collected data included an examination of the reliability of the obtained answers (variables) using Cronbach's alpha coefficient. In addition, a simple regression analysis was used to examine the four dimensions of sustainability and the impact of SuT on the SoR [57]. The indices include each dimension of sustainability, which in the statistical analysis represent independent variables [55,58,59]. Table 1 shows the obtained values for both groups of respondents.

**Table 1.** Confirmatory factor analysis of each sustainability dimension ($n$ = 630).

| Items | Residents ($n$ = 400) | | | | Visitors ($n$ = 230) | | | |
|---|---|---|---|---|---|---|---|---|
| **Aspects of SuT** | Loading | ($p$-Value) | α | Mean | Loading | ($p$-Value) | α | Mean |
| Institutional Aspects | | | 0.701 | 3.29 | | | 0.749 | 3.36 |
| Trained guides and community representatives escort visitors around the PA | 0.51 | <0.001 | | 2.89 | 0.69 | <0.001 | | 3.02 |
| Local brands (wineries, ethno houses, handmade items, regional businesses, etc.) are evident to visitors in the PA | 0.82 | <0.001 | | 3.14 | 0.71 | <0.001 | | 3.42 |
| The manager's directions for visitor activities and nature preservation are adhered to in the PA | 0.39 | <0.001 | | 3.68 | 0.56 | <0.001 | | 3.49 |
| Information about the history of the reserve, its people, and its communities is available to visitors | 0.63 | <0.001 | | 3.44 | 0.69 | <0.001 | | 3.51 |
| Dimension of Ecology | | | 0.782 | 3.72 | | | 0.759 | 4.13 |
| The protection of the environment is a shared responsibility between locals and visitors | 0.59 | <0.001 | | 4.02 | 0.72 | <0.001 | | 4.34 |
| The PA provides facilities, services, and events that benefit tourists and the local community | 0.67 | <0.001 | | 4.14 | 0.55 | <0.001 | | 4.44 |
| Facilities for tourists exist that do not harm the environment | 0.49 | <0.001 | | 3.01 | 0.52 | <0.001 | | 3.62 |
| Economic Dimension | | | 0.766 | 3.24 | | | 0.812 | 3.51 |
| Residents in the PA gain from tourism | 0.44 | <0.001 | | 2.51 | 0.39 | <0.001 | | 3.11 |
| The PA's tourism industry boosts the regional economy | 0.36 | <0.001 | | 3.02 | 0.42 | <0.001 | | 3.42 |
| An increase in tourism in tge PA keeps locals employed | 0.32 | <0.001 | | 3.12 | 0.51 | <0.001 | | 3.05 |
| Visitors can purchase local goods | 0.41 | <0.001 | | 4.11 | 0.58 | <0.001 | | 3.86 |
| The costs of domestic goods are supported by tourists | 0.55 | <0.001 | | 3.44 | 0.67 | <0.001 | | 4.13 |

**Table 1.** *Cont.*

| Items | Residents (*n* = 400) | | | | Visitors (*n* = 230) | | | |
|---|---|---|---|---|---|---|---|---|
| **Aspects of SuT** | **Loading** | **(*p*-Value)** | **α** | **Mean** | **Loading** | **(*p*-Value)** | **α** | **Mean** |
| Sociocultural Aspects | | | 0.802 | 4.00 | | | 0.862 | 4.10 |
| Crafts and household items are attractive to visitors | 0.78 | <0.001 | | 4.22 | 0.81 | <0.001 | | 4.31 |
| The residents and guests interchange | 0.64 | <0.001 | | 4.14 | 0.63 | <0.001 | | 4.19 |
| Tourists are curious about regional customs and traditions | 0.59 | <0.001 | | 4.05 | 0.41 | <0.001 | | 3.81 |
| Tourists attend local cultural venues and events | 0.69 | <0.001 | | 3.81 | 0.72 | <0.001 | | 4.18 |
| Historical sites pique the interest of visitors | 0.60 | <0.001 | | 3.79 | 0.73 | <0.001 | | 4.01 |

Items measured on a 5-point Likert agreement scale; α—Cronbach's alpha reliability.

By applying the CFA statistical model, it was concluded that all the variables (items) and dimensions of sustainability were supported as valid for analysis and all the *t*-values had statistical significance.

The individual average values for all the items of sustainability, for both groups of respondents, can be seen in Figure 4.

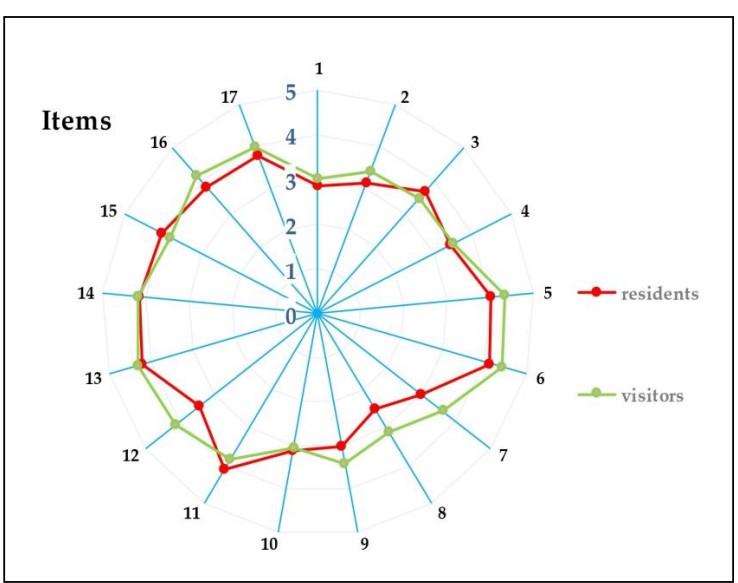

**Figure 4.** Average values for all items for residents and visitors.

For both respondent groups, the overall mean values of satisfaction with the development of SuT were 4.03 and 3.85 (Table 2).

**Table 2.** Scale items for the satisfaction index.

| Items | Residents (*n* = 400) | | | | Visitors (*n* = 230) | | | |
|---|---|---|---|---|---|---|---|---|
| **Index** | **loading** | **(*p*-Value)** | **α** | **Mean** | **loading** | **(*p*-Value)** | **α** | **Mean** |
| | | | 0.701 | 4.03 | | | 0.840 | 3.85 |
| I get a lot of benefits from tourism in this PA | 0.54 | <0.001 | | 2.89 | 0.63 | <0.001 | | 3.02 |
| I'm happy since tourism makes this PA more appealing | 0.54 | <0.001 | | 3.14 | 0.66 | <0.001 | | 3.42 |
| It is crucial that the SnR has tourism | 0.49 | <0.001 | | 3.68 | 0.56 | <0.001 | | 3.49 |
| The level of tourism in this SnR satisfies me | 0.45 | <0.001 | | 3.44 | 0.54 | <0.001 | | 3.51 |

By applying regression analysis to the statistics, the level of SoR with SuT can be determined, and this was applied in this research in such a way that each aspect of sustainability was separately measured and analyzed. The four dimensions of sustainability are examined as part of the statistical analysis with the help of Cronbach's alpha reliability. In this way, it was determined whether the dimensions are reliable for measuring the impact on the respondents' satisfaction ($\alpha \geq 0.60$). Also, the individual values of each dimension were analyzed after applying the regression analysis, which indicates the individual impacts on the respondents' satisfaction [55,60]. Here, the scientific assumption is based on the fact that all four characteristics of sustainability are important markers of tourism satisfaction, accounting for 36% (residents) and 39% (visitors) of the variance ($R_1^2 = 0.362$; $R_2^2 = 0.393$) (Table 3).

**Table 3.** Regression analysis of SoR ($n = 630$).

| Satisfaction of Tourism | Residents | | Visitors | |
|---|---|---|---|---|
| | β1 | *p*-Value | β1 | *p*-Value |
| Institutional dimension | 0.184 | 0.010 | 0.211 | 0.102 |
| Ecological dimension | 0.277 | 0.031 | 0.251 | 0.111 |
| Economic dimension | 0.156 | 0.026 | 0.201 | 0.137 |
| Sociocultural dimension | 0.254 | 0.152 | 0.233 | 0.159 |

Standardized β value used $R_1^2 = 0.362$; $R_2^2 = 0.393$.

## 6. Discussion

The institutional aspect of sustainability had the lowest mean value for each of the two response groups (3.29 and 3.36). Community members and visitors gave the least credence to the claim that trained guides and community representatives escort visitors around the PA. The values of the institutional dimension of sustainability indicated the importance of these factors to all participants and subjects of TD. In the process of planning the development of tourism and creating planning measures, a more significant role of the local population is necessary. This activity can be manifested through different nature classes with the basic goal of educating visitors about local crafts and brands and the way certain domestic products are produced. In addition, this educational form of tourism can also include the expansion of knowledge about the protection of ecosystems in this PA. By implementing various activities and strengthening the role of the local population, the institutional dimension of sustainability and the factors that condition it can be positively influenced.

The sociocultural dimension (4.00 and 4.10) and the average values of the ecological characteristics of sustainability (3.72 and 4.13) are noticeably higher. Through their analysis, it may be said that the mentioned indicators possess a stronger effect on SuT. Also, these two dimensions of sustainability have pronounced factors that are catalysts for the development of tourism within this SnR. Forms of tourism that would be compatible with nature and its values in this reserve would be nature-based forms of tourism, like environmental tourism, scientific research tourism, observing birds, taking pictures of the environment, etc. Sociocultural sustainability as an important pillar of SuT implies the growth of event, cultural, wine, adventure, gastronomic, and alternative types of tourism that emphasize the rich ethno-social values of the local population. The sociocultural aspect of tourism is also based on strengthening the interaction between visitors and the local community. The inclusion of PAs in the tourist offering is not possible without the clearly defined role of the local community in SuT. The local population is considered an important creator of defining the tourist product [15,61]. The interaction between the local population, the managers of the PA, and visitors is the basic prerequisite for SuT [55]. During the development of tourist infrastructure in TB, a more significant role is needed of visitors attending educational centers, as well as guide services, in which the representatives of the local community are the bearers of education. Proper TD also concerns the construction of facilities for the reception of tourists and the strengthening of ethnic settlements in rural areas [62–65].

Buildings and infrastructure must not have a negative impact on the environment [66,67], and the level of construction must be in line with the carrying capacity [39] and protection zones of this PA [68].

Analyzing the impact of SuT on the SoR, relatively the same values are observed (Table 2). The overall mean value of satisfaction with the development of SuT for residents and visitors is 4.03 and 3.85. Cronbach's alpha values of 0.70 and 0.84 indicate the reliability of these variables for analysis. Residents and visitors are of the opinion that it is important to develop tourism in this SnR and that these activities can result in various benefits, both for residents and visitors. When preparing planning documents and TD studies, it is necessary to emphasize those forms of tourist activities that bring residents and visitors into direct contact [30,69]. During joint contact, awareness of the importance of the PA in the development of SuT and the need to protect the space and preserve natural values is strengthened [33,70–74].

The results of applying the regression analysis (Table 3) indicate that SuT affects the SoR, according to the four aspects of sustainability ($0.010 > p > 0.159$). These values are the result of the impact of the dimensions of sustainability on the overall growth of tourism in TB. Each of the dimensions, to a certain extent, influences the SoR. The main objective of this research is to determine whether SuT affects the SoR. After analyzing the obtained data, it can be concluded that SuT significantly affects the SoR. One of the significant results of this research is that when planning and developing tourism in TB, the role of residents in all tourism activities must be strengthened. The local population is an important pillar of SuT. Also, this SnR can be an important integral part of the local, regional, and national tourism levels and can play an important role in the SuTD [75,76].

The results of the theoretical analysis reveal the conclusions that PAs are destinations that can have an important function in SuT. The ecological and institutional aspects of sustainability are recognized by locals as the most important dimensions. By strengthening the ecological and institutional values of these destinations and the proper development of tourism, economic income from tourism and potential jobs can be provided, which is important for local families. In this way, the functioning of the invisible but tangible circular system of sustainable development can be ensured. If the results of the theoretical analysis are compared with the results of this research, a significant coincidence can be noted. In this research, ecological and sociocultural sustainability were recognized by the residents as the most important features to the development of tourism. Low values for the institutional dimension of the results reflect the absence of the implementation of various planning measures of development and control. Insufficient training of the locals and inadequate involvement of the local population in the planning and development of sustainable forms of tourism also influenced the lower values. An important factor is also the absence of sufficient financing of various projects by the state and its authorities, which can sometimes have significant consequences.

A significant theoretical conclusion is that when planning the development of tourism within PA, special attention must be paid to the role of the population. The research's theoretical contribution originates from the observation that there are important prerequisites for the growth of SuT if the four sustainability dimensions account for a roughly equal proportion of the conditions of sustainable tourism. The practical contribution of this research indicates the importance of the role of the local community in sustainable development. The direct role of the local population in the development of the destination and tourism is of key importance, through control over development progress, education, the promotion of local products, and strengthening interactions with visitors. These data can be important in the creation of national tourism development strategies and in the strengthening of underdeveloped areas, where tourism can be the primary economic area. Also, the results of the research imply the importance of the growth of local crafts, educational centers, schools in nature, the promotion of domestic products, culture, and gastronomy, which are important for both residents and visitors of PAs.

## 7. Conclusions

TB can be an important tourist destination because it has a wealth of natural and social tourist motives, which have not yet been sufficiently valorized and integrated into the tourist offering. The foundation of this PA's tourism development is the preservation of the ecological and social values of this sensitive destination, the abolition of tourism's harmful effects, the augmentation of the participation of locals and tourists in the planning and development of tourism, the creation of unique forms of tourism, and other initiatives. The successful implementation of ecological components, environmental and ecosystem protection, and giving the advantage to tourist activities organized in accordance with ecological principles are imperative to the implementation of marketing the tourist activities of this PA. From the aspect of SuT, it is also important to highlight the intangible cultural heritage of the local population that lives around this PA, i.e., the tourist destination. This heritage can be successfully marketed in the form of a tourist offering, and it has the task of introducing visitors, to the greatest extent, to culture, tradition, customs, folklore, ways of eating, cultural manifestations, and many other ethnic– social tourist motives.

Because SuT has a positive impact on total economic development, it is one of the most frequently recognized kinds of space utilization [77–79]. Through the activation and promotion of PAs as attractive destinations, only recently has tourism emerged as a catalyst for these activities. SuTD is specifically a model for organizing various tourism activities that need to adhere to the requirements of having a beneficial effect on the environment [80]. Environmental, sociocultural, and economic impacts stand out as the most significant impacts of SuT [81–84]. SuTD in this PA should include a set of planning activities and protection measures, with the basic goal of improving natural and social values.

By applying the PoS model in this research, the result was that TB was rated, by its residents and visitors, as an important tourist destination for SuT. Comprehensive research has indicated this conclusion in the responses related to the perception of residents and visitors of certain claims and their satisfaction with the dimensions of sustainability in this SnR. If the individual average values of the responses are compared, a relatively small difference can be observed in the values of the responses of the residents and visitors, observed across all four dimensions. Both groups of respondents provided the lowest rated values for the institutional dimension of sustainability. The factors that define this dimension concern the management of the PA, legislators, and facilities providing different information [85–87]. By strengthening the institutions that directly or indirectly determine the protection and management of this SnR, an increase in the institutional sustainability factors can be brought about. Furthermore, it is imperative to enhance the involvement of community leaders in the models of tourism preparation, promotion, growth, and governance within this PA. It can be said that the prerequisites for defining SuT have been met if the growth of tourism benefits TB ecologically, economically, sociologically, and on an institutional basis [88,89].

Serious limitations of the research were the lack of competence among the individual respondents in understanding and correctly responding to the claims regarding sustainable tourism and the dimensions of sustainability. This is why it is important to include an introductory part of the questionnaire and provide oral support when filling out the survey form, which made the examination procedure significantly more difficult. The authors will concentrate their future research on the study of the state and perspectives of the development of sustainable tourism in other protected areas in Vojvodina. The territory of the province has a significant number of PAs that differ in terms of the structure of the living world in them and their geology, ecosystems, and spatial distribution. By researching SuT in a large number of research areas, more reliable results of national significance can be obtained. Apart from that, the authors will focus their future research on examination of the possibilities for the development of SuT in the PAs of the countries in this region and of the world, which will be useful for a comparative analysis with the already obtained results of the examination of SuT in the selected PAs of the AP of Vojvodina.

**Author Contributions:** Conceptualization, I.T., S.Š., A.N.C., F.N., M.A., M.P. (Marko Pavlović), T.S. and M.P. (Marija Perić), methodology, I.T., A.N.C., F.N., M.A., T.S. and M.P. (Marija Perić); software, I.T., S.Š., F.N., M.A., M.P. (Marko Pavlović), T.S. and M.P. (Marija Perić); validation, I.T., S.Š., A.N.C., F.N., M.A., M.P. (Marko Pavlović), and T.S.; formal analysis, I.T., S.Š., A.N.C., F.N., M.A., T.S. and M.P. (Marija Perić); investigation, I.T., S.Š., A.N.C., F.N., M.A., M.P. (Marko Pavlović), and M.P. (Marija Perić); resources, I.T., S.Š., A.N.C., M.A., M.P. (Marko Pavlović), T.S. and M.P. (Marija Perić); data curation, I.T., A.N.C., F.N., M.A., M.P. (Marko Pavlović), T.S. and M.P. (Marija Perić); writing—original draft preparation, I.T., S.Š., A.N.C., F.N., M.A., M.P. (Marko Pavlović), T.S. and M.P. (Marija Perić); writing—review and editing, I.T., S.Š., A.N.C., F.N., M.A., T.S. and M.P. (Marija Perić); visualization, I.T., S.Š., A.N.C., F.N., M.A., M.P. (Marko Pavlović), T.S. and M.P. (Marija Perić); supervision, I.T., S.Š., A.N.C., F.N., M.A., M.P. (Marko Pavlović), T.S. and M.P. (Marija Perić); project administration, I.T., S.Š., A.N.C., F.N., M.P. (Marko Pavlović), T.S. and M.P. (Marija Perić) funding acquisition, I.T., S.Š., A.N.C., M.A., M.P. (Marko Pavlović), T.S. and M.P. (Marija Perić). All authors have read and agreed to the published version of the manuscript.

**Funding:** This research received no external funding.

**Institutional Review Board Statement:** Not applicable.

**Informed Consent Statement:** Not applicable.

**Data Availability Statement:** The data that support the findings of this study are available upon reasonable request from the authors.

**Conflicts of Interest:** The authors declare no conflicts of interest.

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
