# Peer review of "The Impact of Sustainable Tourism on Resident and Visitor Satisfaction—The Case of the Special Nature Reserve “Titelski Breg”, Vojvodina"

_sustainability, doi:10.3390/su16072720_

Round 1

Reviewer 1 Report

Comments and Suggestions for Authors

Dear authors thank you for providing me with interesting material and read titled The impact of sustainable tourism on the residents and visitors satisfaction – a case of Special Nature Reserve "Titelski Breg", Vojvodina. Authors used quantitative approach in measuring state of sustainable tourism and its impact on respondents’ level of satisfaction. To my understanding they used sustainability scale previously developed and collected data with two sample groups residents and visitors. After which they measured impact of sustainability dimensions on visitor and resident satisfaction through simple regression analysis.

Even if the research is interesting and well presented, I have some questions and suggestions for improvement on certain parts of the manuscript.

1.        In the introduction part I would suggest including literature that is connected to the Titel Loess Plateau and geotourism. Even if area is well known for eco-tourism value, its value also lies in geotourism.

More about geoconservation and geotourism potential can be found in the work of:

Vasiljević, Dj.A., Marković, S.B., Hose, T.A., Ding, Z., Guo, Z., Liu, X., Smalley, I., Lukić, T., Vujičić., M.D. (2014): Loess–palaeosol sequences in China and Europe: Common values and geoconservation issues. Catena 117, 108-118.

Also, I find sentence “Limitations in this research concerned the impossibility of making certain personal contacts with residents and visitors due to significant weather disasters, which often affected this region of AP Vojvodina during the period of surveying respondents, that is, fieldwork and data collection.” quite difficult to follow, I would recommend rephrasing. But, even with that, I don’t understand what authors want to say, that it was difficult to get in personal contact with the respondents due to weather conditions? Why not conduct the research in different time of the year, it’s not like it’s not reachable during most time of the year.

What I lack the most in the introduction part and what should be better elaborated in the research part is the research gap, from which research questions should be derived. I would suggest including this in the introduction section.

2.        My main concern is about the methodology and contribution to the research body of knowledge. I presume that authors took already developed instrument, validated scale for sustainable tourism and in order to test scale reliability they used Cronbach alpha. Manuscript would really benefit if the authors conducted EFA and CFA for sustainability development scale. I won’t suggest authors to conduct additional research direction, but I think that including more indices that affect scale reliability and discriminant validity of the construct would be beneficial to the manuscript. I suggest including Composite reliability (CR) and Average variance extracted (AVE) indices for scale reliability and for discriminant validity including Fornell-Larcker criterion and heterotrait-monotrait ratio (HTMT). This would contribute to the measurement model is dependable.

Also, I would suggest restructuring methodology part a bit, to include subsections: Procedure, Instruments, Data Analysis as in that way it would be more beneficial to the reader and more easier to follow.

3.        Within discussion part I would suggest excluding the part about the Cronbach alpha and reliability of the scale, as this text should be, and is already, within results part. Discussion should be about dimensions of the construct and impact on satisfaction.

4.        In the Conclusion part, I would suggest adding sections about limitations to the research and also further research and research directions.

Good luck with the manuscript and further research

Author Response

Dear Sir or Madam,

We are grateful for all the recommendations for improving the manuscript. The changes made our article gain scientific quality. We inform you about changes and additions:

  • In the introduction part I would suggest including literature that is connected to the Titel Loess Plateau and geotourism. Even if area is well known for eco-tourism value, its value also lies in geotourism.

Answer: We added crucial details regarding the loess section and its significance for geotourism to the Introduction chapter.

  • Also, I find sentence “Limitations in this research concerned the impossibility of making certain personal contacts with residents and visitors due to significant weather disasters, which often affected this region of AP Vojvodina during the period of surveying respondents, that is, fieldwork and data collection.” quite difficult to follow, I would recommend rephrasing. But, even with that, I don’t understand what authors want to say, that it was difficult to get in personal contact with the respondents due to weather conditions? Why not conduct the research in different time of the year, it’s not like it’s not reachable during most time of the year.

Answer: The sentence has been modified so that all readers may understand it. We appreciate you bringing out this mistake. Lines 111–116.

  • What I lack the most in the introduction part and what should be better elaborated in the research part is the research gap, from which research questions should be derived. I would suggest including this in the introduction section.

Answer: We appreciate your idea and would like to let you know that the introduction chapter has been updated. Lines 75–87.

  • My main concern is about the methodology and contribution to the research body of knowledge. I presume that authors took already developed instrument, validated scale for sustainable tourism and in order to test scale reliability they used Cronbach alpha. Manuscript would really benefit if the authors conducted EFA and CFA for sustainability development scale. I won’t suggest authors to conduct additional research direction, but I think that including more indices that affect scale reliability and discriminant validity of the construct would be beneficial to the manuscript. I suggest including Composite reliability (CR) and Average variance extracted (AVE) indices for scale reliability and for discriminant validity including Fornell-Larcker criterion and heterotrait-monotrait ratio (HTMT). This would contribute to the measurement model is dependable.

Answer: We have considered your proposal in detail, and we want to thank you for it. This research uses a standardized questionnaire that is part of the PoS research model. The research model has been used so far in examining sustainable tourism in different destinations, many times and by different researchers. The authors used this model in over 20 studies. This study is an extension of the author's earlier research, as reported in the paper. So, it is going to be integrated later into a single, large survey. One segment of statistical data processing involves determining the validity of each of the four dimensions of sustainability, based on the respondents' answers. In identical research from a long time ago, the authors Cortina [1993] and Nunnally and Bernstein [1994] suggest that an alpha value of less than 0.60 can be accepted in research studies. That's what we wrote in the text (line: 319). This postulate is still used today in similar and identical research. Using the alpha coefficient in this research, α values of over 0.7 and 0.8 were determined (Tables 1 and 2). According to the mentioned sources, it is considered a very reliable coefficient, which is why the variables can be taken into consideration with certainty. Therefore, we are of the opinion that the text should not be burdened with additional value checks. That would perhaps detract a little from the adopted research model.

  • Also, I would suggest restructuring methodology part a bit, to include subsections: Procedure, Instruments, Data Analysis as in that way it would be more beneficial to the reader and more easier to follow.

Answer: We would like to inform you that we have divided the Methodology chapter according to your suggestion. Lines: 268-334.

  • Within discussion part I would suggest excluding the part about the Cronbach alpha and reliability of the scale, as this text should be, and is already, within results part.

Answer: We shortened the text according to the suggestion. Lines: 379-389.In the Conclusion part, I would suggest adding sections about limitations to the research and also further research and research directions.

Answer: We appreciate this suggestion and inform you that we have expanded the Conclusion chapter for additional information. Lines: 535-545.

Thank you again for your efforts.

With all our respect

Authors

Reviewer 2 Report

Comments and Suggestions for Authors

The methodology did n`t bring something new, but is well applied. The area of research has local interest for readers, so more similar international studies should be a base of comparation for authors.

Author Response

Respected reviewer,

We appreciate your confidence in us. In response to your recommendation, we made the text longer. The modifications improved the article's scientific quality.

We let you know about additions and modifications:

The area of research has local interest for readers, so more similar international studies should be a base of comparation for authors.

Answer: We expanded the Literature review chapter with additional research results, lines: 172-222.

Thank you again for your efforts.

With all our respect

Authors

Reviewer 3 Report

Comments and Suggestions for Authors

Abstract: The background of the study is elaborately introduced, which is commendable. However, the presentation of the research findings is too brief, mentioning only the relationship between SuT and SoR without detailing the nature of this relationship. The abstract should conclude with the implications of the study rather than merely presenting results.

Literature Review (LR), Line 86: "The SnRs of Vojvodina differ significantly in terms of ecosystem, landscape, geological, and biological characteristics." The purpose of this statement is unclear. It is recommended to briefly introduce where these differences lie.

Lines 124-146: These paragraphs mainly introduce two related case studies in great detail. While thorough, this may not be the most appropriate expression for a literature review, as two cases are insufficient. It is suggested to categorize and review related research instead.

Lines 202-203: The four dimensions of sustainability are mentioned without citing reliable sources. It is necessary to provide references for these dimensions.

Lines 210-220: "The investigated institutional indicators are the importance of legal legislation for the protection of the area, the importance of management processes, the existence of guiding and educational centers, the possibility of developing scientific forms of tourism, and others." Are these indicators based on previous studies or certain existing theories? Please clarify and cite appropriate research.

Results:

You mentioned the collection of 630 questionnaire responses. Are all these data valid? Was there any method employed to verify the validity of these responses?

It is unlikely that all data could be considered valid without any form of validation, or is the number 630 the result after such validation? Additionally, in Line 262, the execution of the regression analysis lacks detail. More information is needed here.

Discussion:

You highlight the importance of linking theoretical constructs with empirical data more effectively, particularly in the context of sustainability in tourism (SuT).

While the analysis of Cronbach Alpha values and the exploration of various dimensions' impact on SuT are noteworthy, there's a significant opportunity to strengthen the theoretical and empirical nexus. Specifically, comparing theoretical expectations with the study's findings could elucidate this research's contributions or challenges to existing theories.

Additionally, the discussion around low mean values in institutional dimensions warrants a deeper investigation into underlying causes. Questions arise whether these are due to inadequate training among local guides and community representatives, or broader issues in tourism planning and management. It needs to be discussed more.

The paper lacks implications and significance for both theoretical and empirical research. It is advised to include these aspects.

Comments on the Quality of English Language

NA.

Author Response

Respected reviewer,

We appreciate all of the suggestions you made to make the manuscript better. The modifications improved the article's scientific quality.

We let you know about additions and modifications:

  • Abstract: The background of the study is elaborately introduced, which is commendable. However, the presentation of the research findings is too brief, mentioning only the relationship between SuT and SoR without detailing the nature of this relationship. The abstract should conclude with the implications of the study rather than merely presenting results.

Answer: In response to your recommendation, we added material to lines 40–43 of the abstract.

  • Literature Review (LR), Line 86: "The SnRs of Vojvodina differ significantly in terms of ecosystem, landscape, geological, and biological characteristics." The purpose of this statement is unclear. It is recommended to briefly introduce where these differences lie.

Answer: The sentence has been modified so that all readers may understand it. I appreciate you bringing out this mistake. Lines 125–128.

  • Lines 124-146: These paragraphs mainly introduce two related case studies in great detail. While thorough, this may not be the most appropriate expression for a literature review, as two cases are insufficient. It is suggested to categorize and review related research instead.

Answer: We took your advice and revised and enlarged this paragraph for further study. Lines 169–222.

Lines 202-203: The four dimensions of sustainability are mentioned without citing reliable sources. It is necessary to provide references for these dimensions.

Answer: We have included the necessary references in the text. The sentence has been slightly reformulated. Lines: 290-291.

  • "The investigated institutional indicators are the importance of legal legislation for the protection of the area, the importance of management processes, the existence of guiding and educational centers, the possibility of developing scientific forms of tourism, and others." Are these indicators based on previous studies or certain existing theories? Please clarify and cite appropriate research.

Answer: We would like to inform you that we have added the necessary references in the text. Lines: 322-334.

  • You mentioned the collection of 630 questionnaire responses. Are all these data valid? Was there any method employed to verify the validity of these responses? It is unlikely that all data could be considered valid without any form of validation, or is the number 630 the result after such validation? Additionally, in Line 262, the execution of the regression analysis lacks detail. More information is needed here.

Answer: We wrote in the text that each questionnaire was controlled in detail by hand, while the validity of the given answers (variables) was tested with the help of the Crombach alpha coefficient. Also, we expanded the text to explain the valid number of questionnaires. Lines: 337-339; 368-374.While the analysis of Cronbach Alpha values and the exploration of various dimensions' impact on SuT are noteworthy, there's a significant opportunity to strengthen the theoretical and empirical nexus. Specifically, comparing theoretical expectations with the study's findings could elucidate this research's contributions or challenges to existing theories.

Answer: We appreciate your input and inform you that we have expanded the text for additional information. Lines: 468-482.

  • The discussion around low mean values in institutional dimensions warrants a deeper investigation into underlying causes. Questions arise whether these are due to inadequate training among local guides and community representatives, or broader issues in tourism planning and management. It needs to be discussed more.

Answer: Thank you for this suggestion. We inform you that we have expanded the Discussion section for the necessary information. Lines: 452-467.

  • The paper lacks implications and significance for both theoretical and empirical research. It is advised to include these aspects.

Answer: In response to your suggestion, we expanded the Conclusion chapter. Lines: 535–549.

Thank you again for your efforts.

With all our respect

Authors

Round 2

Reviewer 1 Report

Comments and Suggestions for Authors

Dear authors,

Cronbach's alpha is a measure used to assess the reliability, or internal consistency of a construct. Even if its most used measure, it's not enough. This is why I asked you to include additional measures to contribute to the construct validity and reliability.

Author Response

Dear Reviewer,

We are extremely pleased that you pointed out the possibilities for improving our work, which we implemented. You also suggested using the Fornell-Larcker criterion and heterotrait-monotrait ratio (HTMT).

In Tables 1 and 2, we have added a report after applying the CFA method of checking the reliability of the variables. We have also added a description of the method to the text (lines: 321-324; 360-362).

Once again, we thank you for your hard work and for taking the time to make our work of the highest quality.

Best Regards

 Authors

Reviewer 3 Report

Comments and Suggestions for Authors

Thanks for your hard work. Most of the concerns have been addressed. 

Comments on the Quality of English Language

Moderate editing of the English language is required.

Author Response

Dear Reviewer,

We thank you for your trust again. We inform you that we have performed additional English proofreading.

Best regards,

Authors